# *UNBRANCHED3* Expression and Inflorescence Development is Mediated by *UNBRANCHED2* and the Distal Enhancer, *KRN4*, in Maize

**Yanfang Du**, **Lei Liu**, **Yong Peng**, **Manfei Li**, **Yunfu Li**, **Dan Liu**, **Xingwang Li**, **Zuxin Zhang***

National Key Laboratory of Crop Genetic Improvement, College of Plant Science and Technology, Huazhong Agricultural University, Wuhan 430070, P.R. China

* zuxinzhang@mail.hzau.edu.cn

**Data Availability Statement:** All relevant data are within the manuscript and its Supporting Information files.

## Abstract

Enhancers are *cis*-acting DNA segments with the ability to increase target gene expression. They show high sensitivity to DNase and contain specific DNA elements in an open chromatin state that allows the binding of transcription factors (TFs). While numerous enhancers are annotated in the maize genome, few have been characterized genetically. *KERNEL ROW NUMBER4* (*KRN4*), an intergenic quantitative trait locus for kernel row number, is assumed to be a *cis*-regulatory element of *UNBRANCHED3* (*UB3*), a key inflorescence gene. However, the mechanism by which *KRN4* controls *UB3* expression remains unclear. Here, we found that *KRN4* exhibits an open chromatin state, harboring sequences that showed high enhancer activity toward the *35S* and *UB3* promoters. *KRN4* is bound by UB2-centered transcription complexes and interacts with the *UB3* promoter by three duplex interactions to affect *UB3* expression. Sequence variation at *KRN4* enhances *ub2* and *ub3* mutant ear fasciation. Therefore, we suggest that *KRN4* functions as a distal enhancer of the *UB3* promoter via chromatin interactions and recruitment of UB2-centered transcription complexes for the fine-tuning of *UB3* expression in meristems of ear inflorescences. These results provide evidence that an intergenic region helps to finely tune gene expression, providing a new perspective on the genetic control of quantitative traits.

## Author summary

With the completion of increasing numbers of plant genome sequences and continuous accumulation of multiomics data, numerous regulatory elements are annotated in those intergenic regions containing open chromatin. Enhancers are *cis*-acting DNA elements with the ability to increase target gene expression. They show high sensitivity to DNase and contain specific DNA elements in an open chromatin state that allows the binding of transcription factors. *KERNEL ROW NUMBER4* (*KRN4*) is an intergenic region located downstream of a key inflorescence gene *UNBRANCHED3* (*UB3*). However, the mechanism by which *KRN4* regulates *UB3* expression remains unknown. Here, we showed the genetic interactions between *KRN4* and *UB3* as well as *UNBRANCHED2* (*UB2*) in

**Funding:** This work was supported by the National Natural Science Foundation of China (91935305, 31701431) and the National Key Research and Development Program of China (2016YFD0100404). The funders had no role in study design, data collection and analysis, decision to publish, or preparation of the manuscript.

**Competing interests:** The authors have declared that no competing interests exist.

controlling inflorescence architecture, enhancer activity of *KRN4* toward *UB3* promoters, and *KRN4-* recruited UB2-centered transcription complex for *UB3* transcription. These results provide evidence that an intergenic region helps to finely tune gene expression and quantitative traits.

## Introduction

A large proportion of the plant genome consists of intergenic regions. In the maize (*Zea mays* L.) genome, intergenic regions account for ~85% of the genome [1,2]. In the 1960s, these regions were referred to as "junk DNA," as they were thought to lack biological function [3]. With the sequencing of increasing numbers of plant genomes, the functional characterization of these intergenic regions remains a great challenge [4]. Through the comprehensive use of next-generation sequencing and joint analysis of multiomics data, this mystery is now being solved [5]. For instance, intergenic regions containing open chromatin occupied by DNaseI hypersensitive sites, micrococcal nuclease (MNase) sites, and histone modifications have been identified at the genome-wide level in Arabidopsis, rice and maize [6–8]. These relatively open chromatin structures are characteristic of functionally important regions including recombination breakpoints, enhancers and other possible remote elements of genes that are involved in important biological processes. For example, MNase-hypersensitive regions account for less than 1% of the maize genome but explain approximately 40% of heritable phenotypic variance [8].

A few intergenic regions have been characterized through cloning genes for complex traits. In rice, a 21 kb genomic region, ~5 kb upstream of *qSW5/GW5* is responsible for its expression and for quantitative variation in grain width and weight [9]. In maize, an 853 bp tandem repeat sequence located 100 kb upstream of *BOOSTER1* (*B1*) modulates anthocyanin biosynthesis by regulating *B1* expression [10,11], and a miniature inverted-repeat transposable element insertion in a 70 kb upstream noncoding regulatory element of *ZmRap2.7* regulates *ZmRAP2.7* expression by affecting its epigenetic modification, leading to early flowering time [12,13]. *TEOSINTE BRANCHED1* (*TB1*) is a key domestication gene whose expression level is associated with shoot branching [14]. Variations in the expression of *TB1* alleles are caused by the insertion of two transposable elements (TEs), *Hopscotch* and *Tourist*, in a region ~60 kb upstream of *TB1*. The presence of the proximal *Hopscotch* element causes increased *TB1* expression, whereas the distal region containing *Tourist* represses its expression [15]. Thus, intergenic regions function in the regulation of the transcription of target genes through complex genetic interaction networks involving epigenetic modification [16,17] and play diverse roles in regulating plant architecture, metabolism, growth, development, and domestication.

Noncoding DNA sequences that regulate gene expression can act in two ways: *in cis* or *in trans*. Most intergenic regions in plant genomes consist of TEs, TE-like sequences and other repeats, which are frequently transcribed into intergenic noncoding RNAs. These noncoding RNAs act as *trans*-acting factors to regulate the transcription, translation or DNA modification of their targets [11,18]. The *cis*-regulation of target genes is achieved *via* complex interactions among *cis*-regulatory elements (CREs) that are a key class of noncoding DNA sequences and act to regulate the transcription of a neighboring gene. The interactions among CREs mainly include promoter-promoter, promoter-enhancer and enhancer-enhancer interactions, in a highly ordered three dimensional (3D) chromatin conformation. CREs in noncoding regions are located up- or downstream of their target genes by up to several Mb and function cooperatively with transcription factors (TFs) in an orientation-independent manner [19–21]. A total

of 10,044 intergenic enhancers were predicted to be present in Arabidopsis based on open chromatin signatures [22]. Thousands of intergenic regions have been identified in maize as enhancers through integrating data on DNA methylation, chromatin accessibility and H3K9ac enrichment [23–25].

*KRN4*, an intergenic quantitative trait locus (QTL) for kernel row number, has been mapped to a region ~60 kb downstream of *UNBRANCHED3* (*UB3*), which negatively regulates maize kernel row number [26,27]. The presence of different *KRN4* alleles in near isogenic lines (NILs) results in different levels of *UB3* expression [27]. However, how *KRN4* regulates the expression of *UB3* long-distance remains unknown. In this study, we evaluated the genetic and molecular effect of *KRN4*, and found that *UB3* expression and inflorescence development is finely tuned by an *UNBRANCHED2* centered transcriptional complex and the distal enhancer *KRN4*.

## Results

### Allelic variation of *KRN4* affects *UB3* expression and kernel row number

*UB3*, a member of the SQUAMOSA promoter-binding protein (SBP)-box family of genes, negatively regulates kernel row number and tassel branch number by controlling initiation of reproductive lateral primordia [26]. The *KRN4* locus was found responsible for kernel row number and tightly associated with *UB3* expression level [27].To confirm the role of *KRN4* in maize inflorescences, we analyzed *UB3* expression in two sets of NILs (Fig 1A–1D). In the H21 genetic background, the expression level of *UB3* linked with the $KRN4^{3.1}$ allele was ~3-fold lower than that of *UB3* linked with the $KRN4^{1.9}$ allele ($p = 0.002$), and the kernel row number of the $KRN4^{3.1}$ -*UB3* genotype was approximately two rows greater than that of the $KRN4^{1.9}$

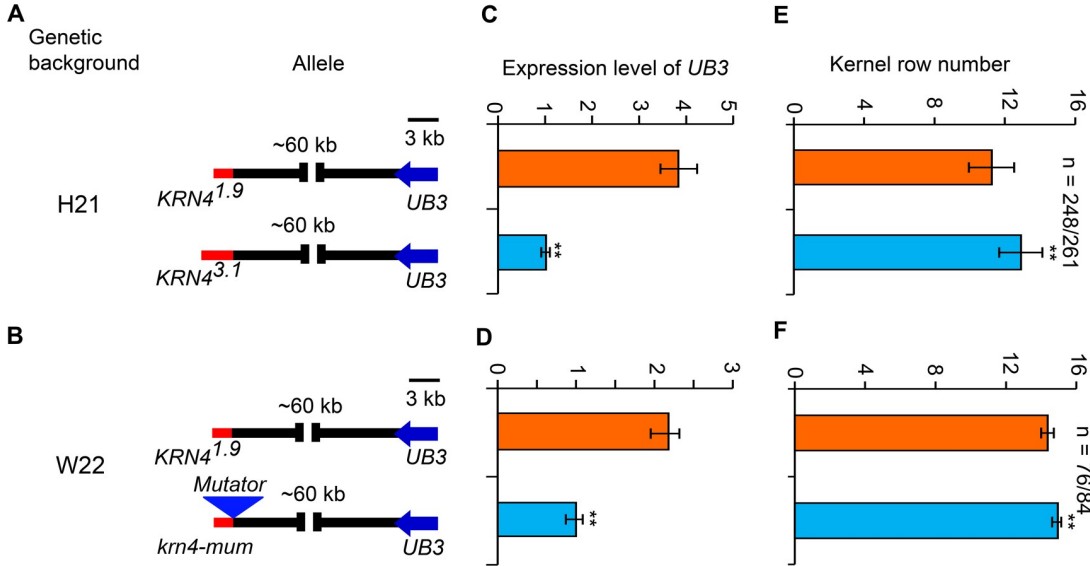

**Fig 1. Genetic effects of *KRN4* alleles in two genetic backgrounds.** (A) Schematic diagrams of *KRN4* alleles ($KRN4^{3.1}$ and $KRN4^{1.9}$) in the H21 genetic background. (B) Schematic diagrams of *KRN4* alleles (*krn4-mum* and $KRN4^{1.9}$) in the W22 genetic background. The red line stands for *KRN4* locus; the triangle indicates the inserted *Mutator* element; bar = 3 kb. (C, D) *UB3* expression levels in 2–3 mm ears of near-isogenic lines in the H21 (C) and W22 (D) genetic backgrounds using quantitative PCR, with three biological replicates in each case. The values show the fold changes. (E, F) Kernel row number of near-isogenic lines in the H21 (E) and W22 (F) genetic backgrounds. Values are shown as the mean ± s.d. Comparsion with the line with $KRN4^{1.9}$, expression level of *UB3* was significantly lower but kernel row number was significantly higher in the line with $KRN^{3.1}$ or *krn-mum*. The *P*-value was calculated by one-way ANOVA. **, p<0.01. n: number of individual ears.

-*UB3* genotype ($p$ = 5.7 ×10$^{-5}$) (Fig 1C and 1E). In the W22 genetic background with the
*KRN4*$^{1.9}$ allele, a *krn4-mum* allele was created by a *Mutator* insertion into the site 55 bp from
the 3′-end of *KRN4* (S1A Fig). The *UB3* expression in *krn4-mum* individuals was lower than
that in *KRN4*$^{1.9}$ homozygotes ($p$ = 0.003), and the kernel row number of *krn4-mum* (14.93
±0.18) individuals was significantly higher than that of *KRN4*$^{1.9}$ individuals (14.36±0.36)
($p$ = 0.03) (Figs 1D and 1F and S1A and S1B). These results demonstrate that *KRN4* functions
as a long-distant regulator of *UB3* expression and thus kernel row number.

## *UB3* regulates inflorescence architecture

Because overexpression of *UB3* strongly represses maize callus regeneration [28], we generated
maize lines showing moderate *UB3* expression by introducing the *UB3* native promoter-driven
*UB3* coding sequence-yellow fluorescent protein (YFP) fusion construct into the inbred line
ZZC01. From the three transgenic lines obtained (transgenic lines with the *UB3* Native Pro-
moter were referred to as UB3-UP2, UB3-UP4, UB3-UP5), *UB3* expression and inflorescence
architecture traits were determined. In addition to kernel row number, we determined tassel
branch number (TBN). We found that in 2–3 mm ears, *UB3* expression in the transgenic lines
was significantly higher than that in the respective non-transgenic lines (UB3-NT2, UB3-NT4,
UB3-NT5) (Fig 2A). The transgenic line UB3-UP4 produced less kernel row number
(14.48 ± 1.22) and TBN (6.43 ± 1.47) than the nontransgenic line UB3-NT4 (15.43 ± 1.18 for
kernel row number and 7.70 ±1.40 for TBN) (Fig 2B–2D). Similarly, reduced kernel row num-
ber, TBN and tassel length (TL) were observed in UB3-UP2 and UB3-UP5 compared with
UB3-NT2 and UB3-NT5, respectively (Figs 2B–2D and S2A), indicating negative regulations
of inflorescence branches and length by *UB3*. Consistent with the results regarding kernel row
number, the diameter of the ear inflorescence meristem (IM) was significantly smaller in
UB3-UP4 (597.2 ± 36.9 μm) than in UB3-NT4 (648.4 ± 30.4 μm) (Fig 2F and 2G). Further-
more, to determine the expression pattern of *UB3* transcripts in the immature ear, we did
mRNA *in situ* hybridization. *UB3* mRNAs were enriched in the peripheral zone of the ear IM
and the spikelet pair meristems (SPMs), but not in the center of the ear IM (Fig 2H and 2I),
similar to the results using a UB3 antibody for immunolocalization [26]. The size of the IM is
thought to be positively correlated with kernel row number [29]; therefore, we propose that
*UB3* regulates branching in inflorescences by modulating the size of IMs.

## *KRN4* genetically interacts with *UB2*

*UB2*, a paralog of *UB3*, also encoding SBP-box transcription factors, plays a redundant role
with *UB3* in controlling the initiation of reproductive axillary meristems. The *ub2;ub3* double
mutant produces fasciated ears with an increased kernel row number compared to the *ub3* and
the *ub2* single mutant [26]. To identify the effect of *KRN4* on the ear architecture of the *ub2*
and *ub3* mutant, *krn4-mum* (referred to as *krn4*) was crossed to the *ub2;ub3* double mutant
and then self-pollinated to segregate single mutants, double mutants and triple mutants. How-
ever, due to the tight linkage between *UB3* and *KRN4*, we could not detect the *krn4;ub3* and
the *krn4;ub2;ub3* genotypes in a segregating population with 577 individuals. To identify the
phenotype of the ear tip, we defined fasciated ears on the basis of the ratio of the widest diame-
ter (d1) to the narrowest diameter (d2) at the tip of the ear, as illustrated in Fig 3A. If d1/d2
≥1.2, the ear considered to be fasciated. The results showed that the ears of the *krn4* mutant
resembled those of the *KRN4;UB2;UB3*, exhibiting a cone-like tip, and a 15.5 ± 1.0 kernel row
number on average, which did not significantly differ from the kernel row number of wild
type (15.0 ± 1.2), only 2% of *krn4* ears showed fasciation (Fig 3B and 3C). Notably, approxi-
mately 27% of *ub3* ears and 47% of *ub2* ears showed fasciation. The proportion of fasciated

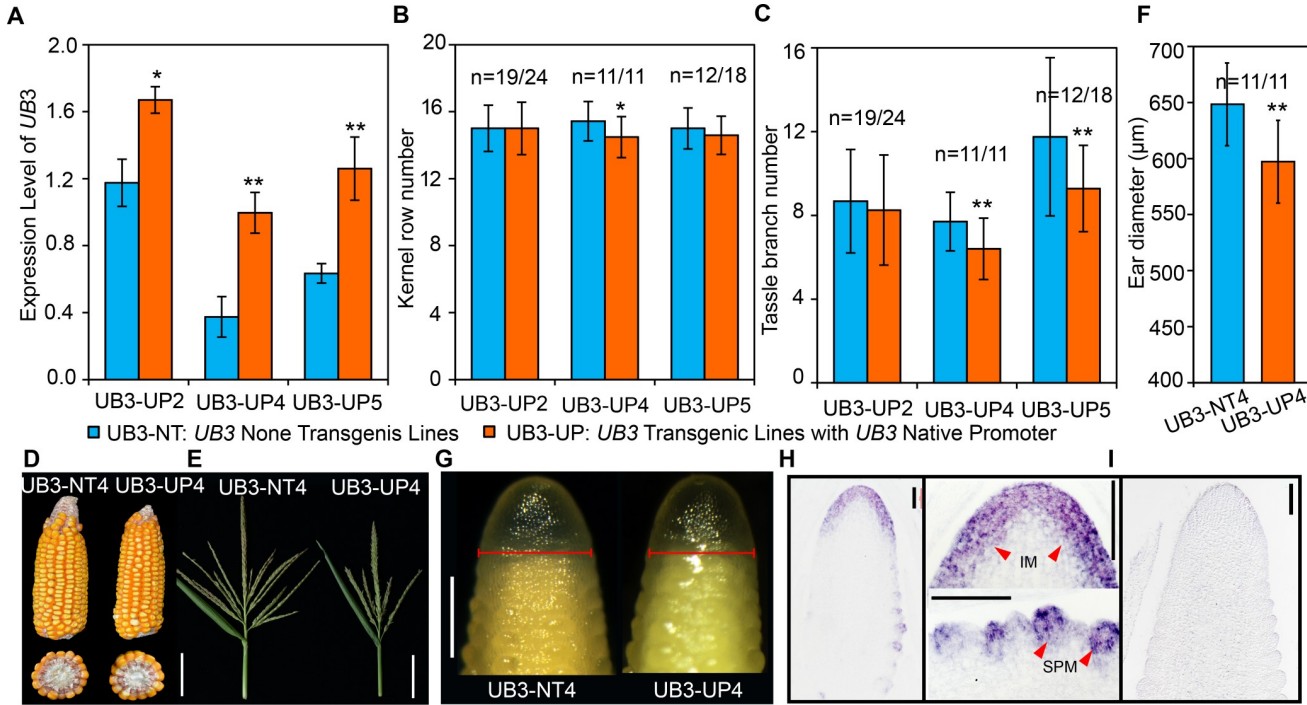

**Fig 2. Inflorescence architecture of *UB3* transgenic lines.** (A) Expression levels of *UB3* in 2–3 mm ears of the three independent transgenic lines and non-transgenic lines, three biological replicates. UB3-UP: *UB3* transgenic lines containing constructs driven by *UB3* native promoter: UB3-UP2, UB3-UP4, UB3-UP5. UB3-NT: *UB3* nontransgenic lines: UB3-NT2, UB3-NT4, UB3-NT5. Values are shown as the mean ± s.e. *, p<0.05; **, p<0.01. (B, C) Phenotypes of inflorescence traits including kernel row number (B), and the tassel branch number (TBN, C) in the *UB3* transgenic and nontransgenic lines. (D-E) Architecture of inflorescences in UB3-NT4 and UB3-UP4. Bar = 6 cm. (F, G) Inflorescence meristems (F) and diameters (G) of 2–3 mm ears in UB3-NT4 (n = 9) and UB3-UP4 (n = 11). Bar = 500 μm in (G). (H, I) mRNA *in situ* hybridization of *UB3* in the inflorescence meristem and spikelet pair meristems (H) using anti-sense probes. The arrow refers to the signal region. *UB3* sense probes as negative controls (I), bar = 100 μm. In (B, C, F) values are shown as the mean ± s.d; n, number of individual ears. The significant differences were estimated using the one-way ANOVA. *, p<0.05; **, p<0.01.

ears was obviously increased in the double mutants: 49% for *ub2;ub3*, and 67% for *krn4;ub2* (Fig 3B and 3C), and the fasciation of the *krn4;ub2* ears was more severe than that in the *ub2* mutant (Fig 3B). These results indicated that *krn4* increases the penetrance and expressivity of fasciated ears in the *krn4;ub2* double mutants, showing genetic interaction between *krn4* and *ub2*. Furthermore, in 2–3 mm ears, *UB3* expression was decreased in both *krn4* and *ub2* mutants relative to that in wild type (Fig 3D). Compared with the expression of *UB3* in *ub2*, that in *krn4;ub2* was lower, showing a coordinated role of both *KRN4* and *UB2* in the regulation of *UB3* transcription (Fig 3D). Additionally, variations at *KRN4* locus strongly influenced *UB3* expression as shown in Fig 1, and *ub2* enhanced defective phenotypes of *ub3* ears as shown in Fig 3B. Thus, we suggest the increased proportion of fasciated ears in *krn4;ub2* is associated with the significant inhibition of *UB3* transcription.

## UB2 is one of the key factors mediating the *KRN4-UB3* interaction

To explore the molecular mechanism of the coordinated roles of both *KRN4* and *UB2*, we generated three transgenic lines (UB2-OE20, UB2-OE21 and UB2-OE23) harboring *pCaMV35S*-driven *UB2-YFP* constructs. In the *ub2* mutant, *UB3* transcription level was 2-fold lower than that in wild type (Fig 4A). In all three transgenic lines, *UB2* transcription levels exhibited >8-fold increases, and *UB3* transcription levels were also increased by 1–2-fold relative to the respective non-transgenic lines (UB2-NTs) (Fig 4B). Additionally, the *UB2*-overexpressing

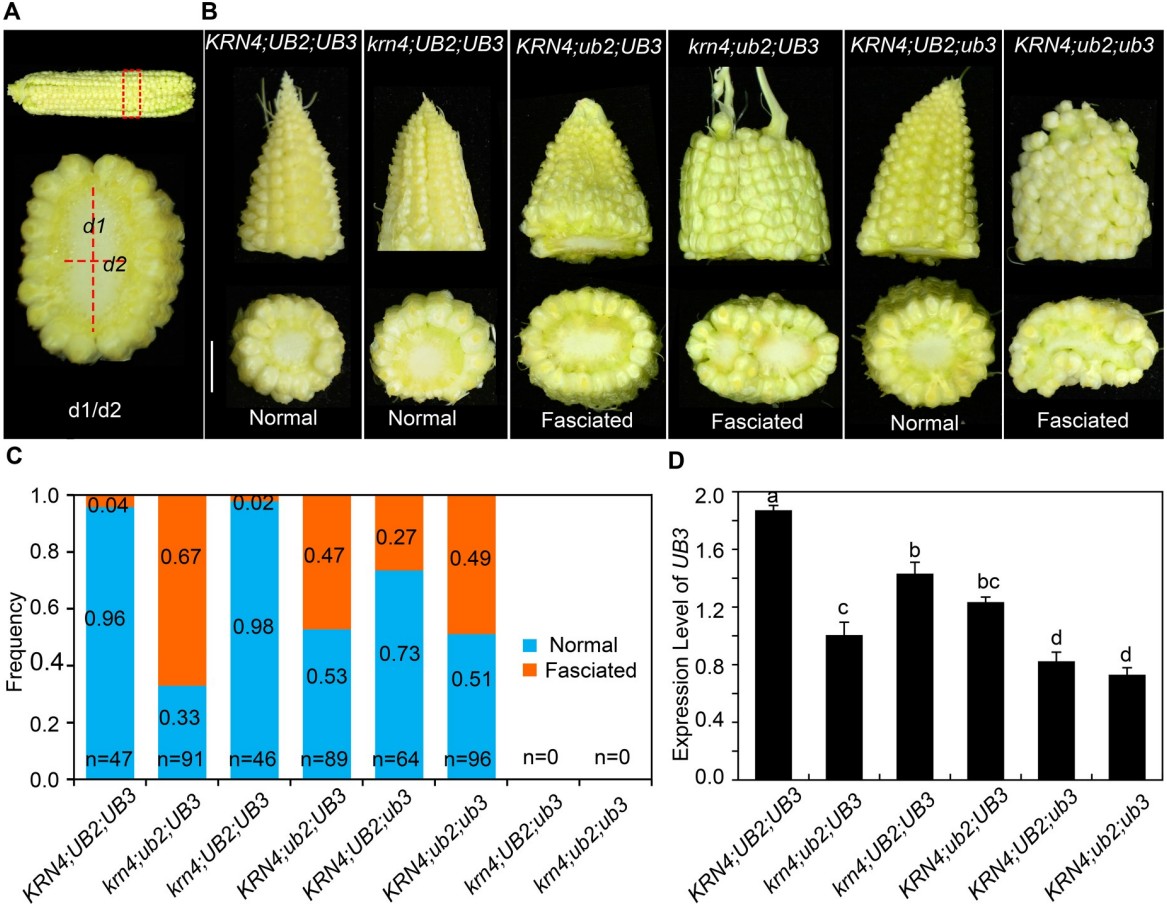

**Fig 3. Genetic interactions between *krn4*, *ub2* and *ub3*.** (A) Definition of the fasciated ears. Both *d1* and *d2* refer to as the widest and narrowest diameters of the ear tip, respectively. (B) Phenotype of ears in the wild type, single and double mutants of *krn4*, *ub2* and *ub3*. Bar = 1 cm. (C) Frequency of fasciated ears in diverse genotypes. n = the number of measured ears. (D) Expression level of *UB3* in 2–3 mm ears of diverse genotypes determined using quantitative PCR, three biological replicates each. Different letter at top of each column indicates a significant difference at P < 0.05 determined by Tukey HSD test.

lines showed significantly reduced kernel row number, TBN and TL relative to the respective UB2-NT lines (Figs 4C and 4D and S2B–S2D). These results suggest that *UB2* might be one of the regulators of *UB3* expression.

To test this hypothesis, we performed chromatin immunoprecipitation sequencing (ChIP-seq) to determine the UB2-occupied targets in the ear inflorescence of UB2-OE20 in which UB2-YFP fusion proteins were detected (S2E and 2F Fig). High-confidence peaks corresponding to UB2-bound DNA regions were found in the *KRN4* region and *UB3* promoter (Fig 4E), indicating the direct binding of UB2 to these two locations *in vivo*, and this binding was confirmed using ChIP-qPCR, which detected >3-fold enrichment in the KRN4-P4 and the UB3-P1 over the input control *in vitro* (Fig 4F), indicating that both *UB3* and *KRN4* are strong candidates for direct targets of UB2.

To further investigate the direct binding of UB2 to *KRN4-P4* and *UB3-P1*, we carefully analyzed the DNA sequences of these fragments and found that they both contain a known plant SQUAMOSA PROMOTER BINDING PROTEIN-LIKE (SPL) specific binding motif (GTAC motif) [30], and then performed an electrophoretic mobility shift assay. We found that UB2 bound to *KRN4-P4* and *UB3-P1 in vitro* (Fig 4G), and detected significantly increased LUC

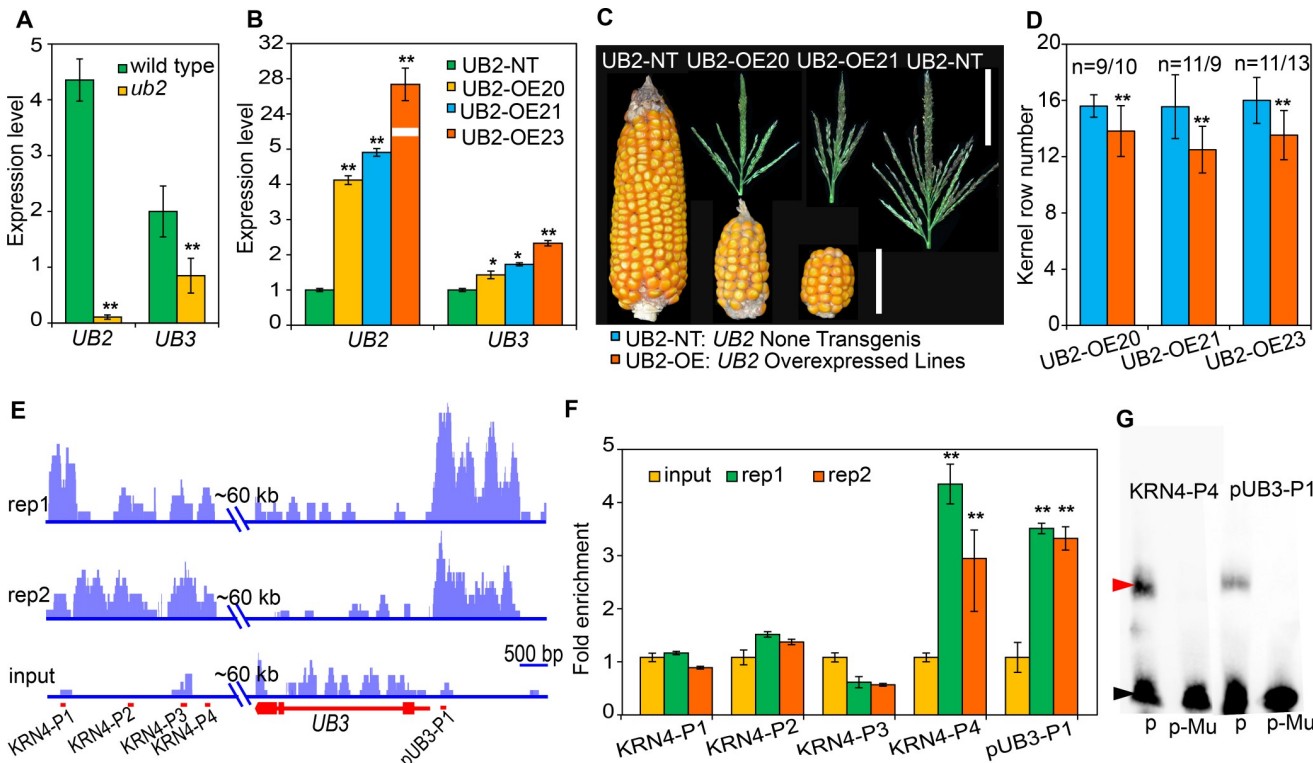

**Fig 4. Expression levels of *UB2* and *UB3* and target genes bound by UB2.** (A, B) Expression levels of *UB2* and *UB3* in 3–5 mm ears of wild type and *ub2* (A), and three independent *UB2*-overexpressing lines and a nontransgenic line (B). Gene expression levels were analyzed by quantitative PCR with three biological replicates. (C) Ears and tassels of *UB2*-overexpressing lines (UB2-OEs) and the nontransgenic line (UB2-NT). Bars = 6 cm for the ears and 10 cm for the tassels. (D) Kernel row number of *UB2*-overexpressing lines and the non-transgenic lines. n, the number of individual ears. The blue and orange columns represent *UB2* non-transgenic and overexpressed lines, respectively. (E) UB2-bound regions in *KRN4* and *UB3* revealed by chromatin immunoprecipitation sequencing (ChIP-seq) with two biological replicates. Bar = 500 bp. (F) Quantitative PCR verification of UB2-bound regions with two biological replicates. The maize tubulin gene (*AC195340.3*) was used as the internal control. (G) Electrophoretic mobility shift assay of UB2-GST fusion protein binding to the KRN4-P4 and pUB3-P1 segments. p, KRN4-P4 or pUB3-P1 probes; p-Mu, mutated probes of KRN4-P4 or pUB3-P1, in which GTAC motif was substituted by GCAC. The upper arrow points to bound probes; the lower arrow points to free probes. The values in (A, B, D and F) are means ± s.d.; *, $p < 0.05$; **, $p < 0.01$, which were estimated by the one-way ANOVA.

activity in cells coexpressing the *p35S-UB2* effector with the *KRN4-pUB3-mp35S-luc* reporter ($p = 3.44 \times 10^{-8}$) compared to cells coexpressing the *p35S-UB2* effector with the *mp35S-luc* reporter (S3 Fig). These results indicated that *UB2* can directly bind to both *KRN4* and the *UB3* promoter to positively regulate *UB3* expression.

### *KRN4* acts as an enhancer spatially interacting with the *UB3* promoter

The 3D configuration of the genome is crucial for the dynamic regulation of gene expression [31]. Chromatin interaction analysis by paired-end tag sequencing (ChIA-PET) is a robust method for capturing genome-wide chromatin interactions [32]. Open chromatin and active enhancers are modified with H3K4me1, H3K4me3 and H3K27ac marks [33,34]. Recently, spatial interactions of the *KRN4* and *UB3* promoter have been detected in apical meristems and seedling leaves [24,25]. These data showed that the spatial interaction anchors identified by H3K4me3 and H3K27ac antibodies occupied three high-confidence remote sites (A1, A2, A3), which were located approximately 33.1 kb—49.9 kb downstream from *UB3* (Fig 5A, data from [24]). In particular, *KRN4* directly interacted with *A1* to form one duplex interaction; *A1* with *A3*, and *A2* with the *UB3* promoter also displayed two duplex interactions (Fig 5A, data from

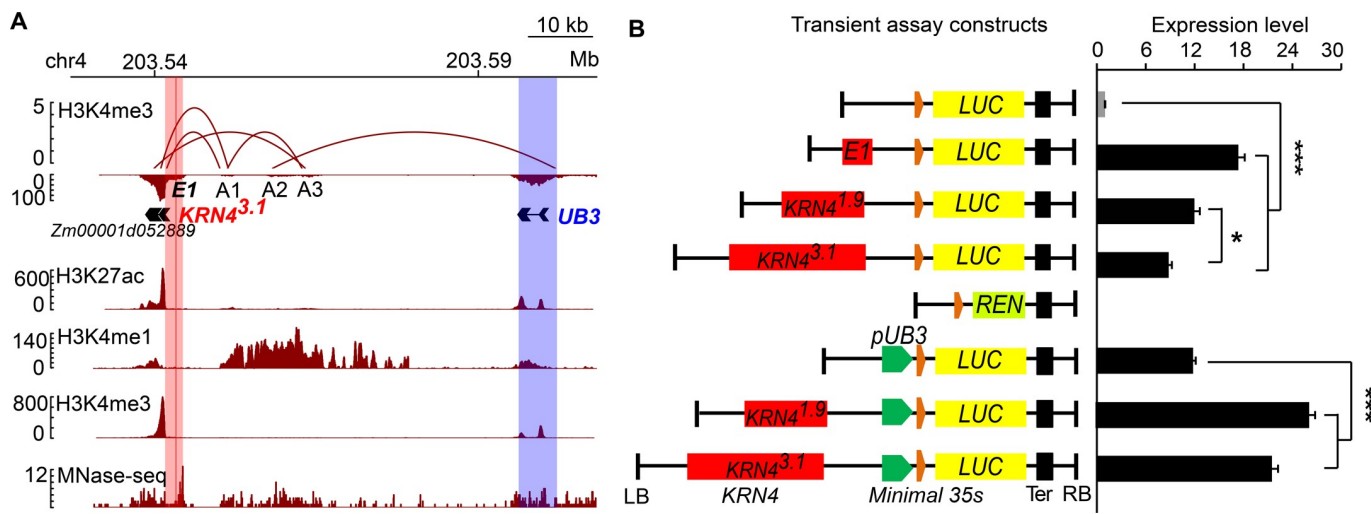

**Fig 5. *KRN4* spatially interacts with *UB3* promoter and acted as an enhancer.** (A) Chromatin interaction analysis between *KRN4* and *UB3* by paired-end tag sequencing (ChIA-PET). Top panel shows interaction loops inferred from H3K4me3 occupancy and the locations of *KRN4^3.1* (light red column), A1, A2, A3 and *UB3* (light blue column). Bottom panels show histone modifications (H3K27ac, H3K4me1, H3K4me3, respectively) and chromatin accessibility (MNase). (B) Transient expression assays showing the effects of *KRN4^3.1*, *KRN4^1.9* and the *E1* segment on gene expression. LB and RB, left and right boundary of the constructs. Orange box, 91 bp minimal *35S*; green box, promoter of *UB3* (*pUB3*); yellow box, *luciferase* ORF (*LUC*); light green box, *Renilla luciferase* ORF (*REN*); black box, nopaline synthase terminator (Ter); red boxes, *KRN4* alleles or 84 bp *E1* element. Transient expression assays were all performed in maize leaf protoplasts. For each transient assay, 4–6 biological replicates and two technical replicates were performed. The value is presented as the mean (LUC/REN) ± s.e. ** $P < 0.01$, * $P < 0.05$.

[24]), showing that the *KRN4-UB3* interaction is mediated by complex interactions among the three duplex interactions. In addition, a strong MNase-hypersensitive signal detected in the *KRN4* region (Fig 5A, data from [8]) indicated an open chromatin state of *KRN4*. ChIA-PET revealed the close proximity between *UB3* and *KRN4* in 3D chromatin configuration, but the spatial proximity between *KRN4* and *UB3* might be indirect due to interconnections among three duplex interactions and the uncertain surrounding regions (S4 Fig). The results, cross-validated by data generated in different laboratories [8,24,25], provided strong direct evidence of the *KRN4-UB3* spatial interaction.

To detect the effect of the spatial *KRN4-UB3* interaction on gene expression, we separately ligated two *KRN4* alleles, *KRN4^3.1* and *KRN4^1.9*, to the minimal 35S promoter (*mp35S*) at a spacing of 2.8 kb (Fig 5B), and then ligated the fusion promoter with the firefly *luciferase* (*LUC*) gene. The transient assays were performed in maize protoplasts to test the effects of *KRN4* alleles on *LUC* expression using *Renilla luciferase* (*REN*) as an internal control. Compared to *mp35S*-driven LUC activity, *KRN4^1.9*-*mp35S*-driven and *KRN4^3.1*-*mp35S*-driven LUC activities were 12-fold ($p = 3.4×10^{-4}$) and 8-fold ($p = 9.3×10^{-5}$) higher, respectively, demonstrating that *LUC* expression was increased by the presence of both *KRN4* alleles, and the enhancement effect of *KRN4^1.9* was greater than that of *KRN4^3.1* (Fig 5B). These findings agree with the *UB3* expression levels detected in both sets of NILs (Fig 1), suggesting that *KRN4* acts as an enhancer to promote *UB3* expression and that the two *KRN4* alleles show differences in enhancer activity. Importantly, both *KRN4* alleles also increased the expression of *UB3* promoter (*pUB3*)-driven *LUC*, and enhancer activity of *KRN4^1.9* was stronger than that of *KRN4^3.1* (Fig 5B).

### *KRN4* exhibits enhancer activity by recruiting the OBF1 and OBF4 proteins

Early studies on enhancers established that sequences with repeated GTGG motifs (where G may be replaced by A) act as enhancers [35]. Within the *KRN4^1.9* region, we detected an 84 bp

fragment harboring five putative enhancer elements (one copy of ATGG, GTAG and GTGA and two copies of GTGG), which was designated *E1* (S5 Fig). The presence of *E1* significantly increased the expression of *mp35S*-driven *LUC*, which was much higher than that driven by *KRN4*[1.9] and *KRN4*[3.1] (Fig 5C). In addition, the 84 bp *E1* region contained an ACGT motif (S5 Fig), which is a specific binding site for basic leucine zipper domain proteins (bZIPs) [36]. We tested the binding of two known maize OCS-binding factors, OBF1 and OBF4 [37,38] to *E1 in vitro* using yeast one-hybrid assays, and revealed both OBF1 and OBF4 bound to *E1* (Fig 6A), their binding was further cross-validated by the presence of OBF1-MBP and OBF4-MBP-DNA interacting bands in the EMSAs (Fig 6B).

To further evaluate the effects of *OBF1* and *OBF4* on the expression of the *E1*-driven downstream genes, we performed transient assays with effector vectors (*CaMV 35S-OBF1* and *CaMV 35S-OBF4*) and reporter vectors (*mp35S-luc* and *E1-mp35S-luc*) (Fig 6C); the *pRL-null* vectors harboring *mp35S-Renlia luciferase* (REN) (*mp35S-REN*) was used as the internal control. Coexpression of *p35S-OBF1* or *p35S-OBF4* with *E1-mp35S-luc* and *mp35S-REN* resulted in a 10- or 9- fold increases in LUC activity, respectively, compared to the control, indicating that both OBF1 and OBF4 target the *E1* to increase *LUC* expression (Fig 6D).

## *UB2*, *UB3*, *OBF1*, and *OBF4* are coexpressed in meristems of the ear inflorescence

Because two enhancer-binding proteins (OBF1 and OBF4) were found to bind to *KRN4* and UB2 also binds to *KRN4* and the *UB3* promoter, we reasoned that these three proteins might

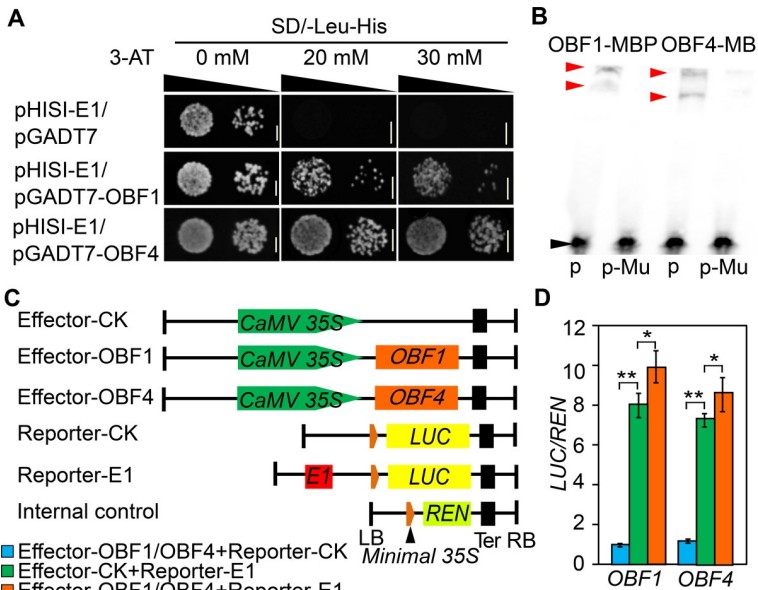

**Fig 6. Enhancer effects of *KRN4* and the binding of OBF1 and OBF4 to the *E1* segment.** (A, B) Yeast one-hybrid and electrophoretic mobility shift assays revealing the binding of OBF1 and OBF4 to the *E1* segment *in vitro*. The upper arrows point to bound probes, and the lower arrow points to free probes. Bar = 0.3 cm. p, E1 probes; p-Mu, mutated E1 probes, in which CACGT motif was substituted by ACATG; (C, D) Transient expression assays showing that the binding of OBF1 and OBF4 to *E1* promotes gene expression. (C) Schematic illustration of the effector and reporter constructs. (D) Luciferase activity detected via transient expression analysis. LB and RB, left and right boundary of the constructs. Orange box, 91 bp minimal *35S*; yellow box, *luciferase* ORF (*LUC*); light green box, *Renilla luciferase* ORF (*REN*); black box, nopaline synthase terminator (Ter); red boxes, 84 bp *E1* element. Transient expression assays were all performed in maize leaf protoplasts. For each transient assay, 4–6 biological replicates and two technical replicates were performed. The value is presented as the mean (LUC/REN) ± s.e. ** $P < 0.01$, * $P < 0.05$.

be recruited by *KRN4* and form a transcriptional complex to fine-tune *UB3* expression. Thus we analyzed the expression patterns of *UB2*, *OBF1*, and *OBF4* in developing ears. The expression domains of *UB2*, *UB3*, *OBF1* and OBF4 in the ear inflorescence partially overlapped. The *UB2*, *UB3*, *OBF1* and *OBF4* transcripts were enriched in the peripheral zone of the IM and SPM primordia, while *UB2* specifically expressed in the basal domain of SPMs (Figs 2H–2I and 7A–7G). The similar spatiotemporal expression patterns of these genes suggest that OBF1, OBF4 and UB2 form a complex and coregulate *UB3* expression.

Furthermore, we fused the coding sequences of *OBF1*, *OBF4* and *UB2* to the C- and N-terminal domains of *LUC* (*CLUC* and *NLUC*, respectively) and used these fusion constructs for luciferase complementation image assays. Cotransfection with *OBF1-CLUC* with *UB2-NLUC* produced weak LUC activity, whereas cotransfection of *OBF4-CLUC* with *OBF1-NLUC* or *UB2-NLUC* produced strong LUC activity (Fig 7H and 7I). By contrast, individual infiltration of the three vectors with the corresponding empty vector failed to produce visible signals, demonstrating a physical interaction between the three proteins. Taken together, the overlapping expression pattern of these four genes and the interactions between OBF1, OBF4 and UB2 proteins suggest the existence of an OBF1, OBF4 and UB2 protein complex in ear inflorescences.

## Discussion

An understanding of the genetic basis of maize inflorescence development has resulted from analysis of a large number of developmental mutants. Among these genes, three SPL genes (*UB2*, *UB3*, *TSH4*) affect inflorescences architecture [26]. Notably, *UB3* is thought to be tightly linked to QTLs for inflorescence traits such as kernel row number and TBN according to analyses in association mapping and linkage mapping populations, and *UB3* genetically interacts with *KRN4* [26,27]. However, it has been unclear how *KRN4* regulates *UB3* expression in meristems of inflorescences. Here, we analyzed the genetic and molecular effects of *KRN4* with *UB2* and *UB3*, to reveal the mechanism whereby *KRN4* fine-tunes *UB3* expression and, thus, inflorescence development.

A large number of functionally important elements (promoters, enhancers, insulators) and transcribed RNAs are being annotated in intergenic regions of the maize genome [1,2]. In a few well-studied cases, intergenic regions have been shown to regulate gene expression and associate with phenotypic variation, such as *B1* [11,12], *TB1* [14], *Hydroxyproline-rich glycoprotein* [39], *Pericarp Color1* [40], *VGT1* [12] and *Benzoxazinless1* (*BX1*) [41]. *KRN4* is also a

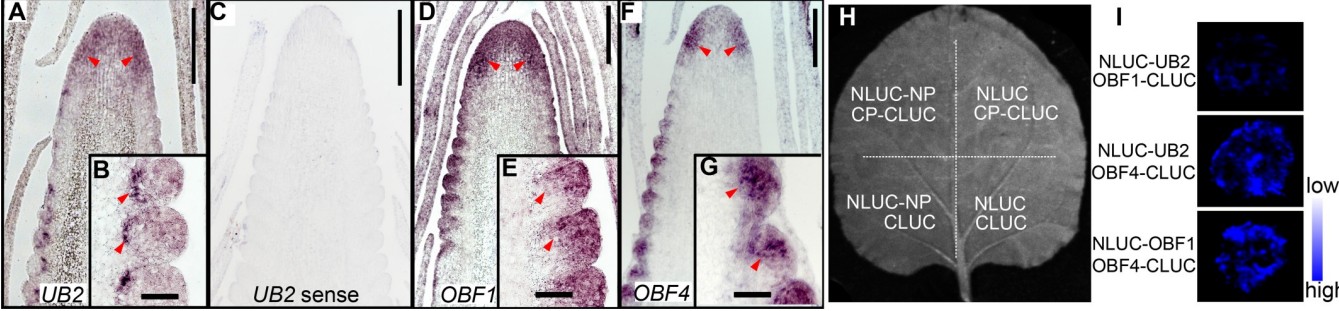

**Fig 7. mRNA expression patterns of *KRN4* binding proteins.** (A-G), mRNA *in situ* hybridization of *UB2* (A-C), *OBF1* (D, E) and *OBF4* (F, G) in 2–3 mm B73 ears using anti-sense probes, (C) *UB2* sense probes as a negative control. The arrows refer to the signal regions. (H, I) Protein–protein interaction identified by a luciferase complementation image assay. (H) Diagram of the experimental design on a single tobacco leaf, including one treatment (upper left) and three control groups. (I) Interaction signals of UB2-OBF1, UB2–OBF4 and OBF1–OBF4. NLUC and CLUC represent N-terminal and C-terminal domains of Firefly Luciferase, respectively. NP and CP, NLUC-fused protein and CLUC-fused protein detected in the luciferase complementation image assay, respectively.

functional intergenic region based on several lines of evidence. First, *KRN4* was mapped to a 3.1 kb intergenic region. Sequence alignment failed to reveal RNA transcripts that were mapped to *KRN4* with high confidence, while a *Harbinger*-like fragment has been mapped to a 1.2 kb insertion/deletion, which causes two allelic variants in NILs [27], suggesting that *KRN4* likely functions as a *cis*-acting element in controlling inflorescence branching. Second, sequence variations in *KRN4* including the 1.2 kb insertion/deletion in the H21 background and a *Mutator* insertion/deletion in the W22 background cause changes in *UB3* expression and quantitative variation of inflorescence branches (kernel row number and tassel branch number). Additionally, genetic interactions between *UB2* and *UB3* [26] and between *KRN4* and *UB3* [27] were found to be involved in the regulation of inflorescence branching. In this study, the double mutant *krn4/ub2* showed an approximately 20-fold higher frequency of fasciated ears than single *krn4* and *ub2* mutants, showing strong genetic interaction between *KRN4* and *UB2* in the regulation of inflorescence branching. Similarly, a genetic interaction between *KRN4* and *UB2* involved in the regulation of *UB3* expression was also found. Finally the *KRN4–UB3* genomic intervals are open chromatin regions, as revealed in the maize MNase HS map [8], and allelic variation of *KRN4* changed *UB3* expression, indicating that the intergenic *KRN4* sequence with open chromatin state achieves its *cis*-activity through spatially interacting with the *UB3* promoter. Two parents used in *KRN4* mapping, H21 which has *KRN4*$^{1.9}$ allele and H21$^{NX531}$ which has *KRN4*$^{3.1}$ allele, show significantly difference in kernel row number and *UB3* expression level [27]. In this study, we found that *KRN4*$^{1.9}$ in H21 exhibits a stronger enhancer activity than *KRN4*$^{3.1}$ in regulating *UB3* expression level that is negatively correlated with kernel row number. The result will help to understand the *KRN4* effects on *UB3* expression level and kernel row number in the original mapping population. This finding also provides an additional case of variation in an intergenic region that fine-tunes gene expression, providing a new perspective on the genetic control of a quantitative trait.

The intergenic QTL *KRN4* has key features often found in enhancers. Intergenic regions usually exert their functions as *cis*-regulatory DNA elements (enhancers or insulators) or *trans*-regulatory factors. Enhancers were initially described as short DNA fragments with the ability to positively drive target gene expression independent of distance and orientation, high sensitivity to DNase, and the presence of specific DNA elements in an open chromatin state that allows the binding of transcription factors (TFs) and transcription coactivators [42,43,44]. In this study, we found that *KRN4* harbored an 84 bp fragment containing five putative enhancer elements. Moreover, the 84 bp fragment and two *KRN4* alleles showed strong transcriptional activation activity toward the promoters of *35S* and *UB3*. Indeed, three TFs including an UB2 and two known enhancer-binding proteins (OBF1 and OBF4), were shown to interact with each other and bind *KRN4*. In particular, *UB2* was able to bind *KRN4* and the *UB3* promoter, and double mutants *ub2;ub3*, *ub2;krn4* showed a higher level of fasciated ears than *ub3* or *krn4* single mutants, suggesting that UB2 is a key mediator of the long-distant regulation of the *UB3* promoter by *KRN4*. One of the most accepted models for long-distance regulatory interactions is the looping model, which hypothesizes that interactions between enhancers–promoters are in physical contact, forming an intervening chromatin loop [45,46]. The chromatin interactions provide close proximity among distant elements. Two recent studies have detected the enriched peaks on *KRN4* locus by H3K4me3, H3K27ac and RNA Pol II [24,25]. The *KRN4–UB3* promoter interaction was mediated by complex interactions among duplexes, while the intervening sequences might be looped outward. Additionally, the *KRN4* region is highly sensitive to MNase, indicating that *KRN4* is in an open chromatin state, which is one of the signatures of an enhancer. These results demonstrated that intergenic *KRN4*, in cooperation with UB2, OBF1 and OBF4, functions as an enhancer of the *UB3* promoter via interconnections among duplex chromatin interactions (S6 Fig).

Since the discovery of enhancer-promoter interactions, the mechanism of their actions is a major interest of the scientific community. The DNA looping and the binding of larger protein complexes explain how enhancers communicate with the targeted promoters over long distances. With the advance of genomics, chromatin immunoprecipitation (ChIP), chromosome conformation capture (3C) and 3C-derived methods combined with sequencing have identified numerous enhancer-like elements in humans and plant species, and have revealed the super-enhancers that are large clusters of highly active enhancers regulating cell type-specific and phenotype-related genes [47,48]. In recent years, a conceptual framework for a phase separation model has been proposed to elucidate transcription control of super enhancers [49]. This model suggests that a high density of proteins and nucleic acids kept in close spatial proximity form phase-separated droplets in order to maintain a compartmentalized and concentrated state for cell type-specific transcriptional regulatory processes [48–50]. Two such super enhancers, BRD4 and MED1 in human cells, control the expression of key genes controlling cell fate and disease occurrence [50]. Whether the transcription factors UB2, OBF1, OBF4 and others enriched in the *KRN4-UB3* chromatin interaction interval form a phase-separated droplet remains to be determined.

## Materials and methods

### Plant materials and phenotypic analysis

The maize *krn4-mum* mutant (mu1023400), a *Mutator* (*Mu*)-insertion mutant, was provided by the Maize Genetics Cooperation Stock Center at the University of Illinois, Champaign-Urbana. The *Mu* is inserted in a 55 bp downstream of the 3'-end of *KRN4*. The *Mu* insertion site was detected by PCR with *KRN4*-specific and TIR6 primers (S2 Table). Heterozygous individuals (+/*krn4-mum*) were selected and crossed to wild-type individuals two times, then self-crossed to segregate the homozygous *krn4-mum* mutants and the wild-type individuals that were evaluated in the field in two environments: summer 2016 in Wuhan (30˚N, 114˚E) and spring 2017 in Sanya (18.34˚N, 109.62˚E), China. Approximately 100 individuals per genotype were evaluated in each environment. One-way ANOVA (analysis of variance) was used to estimate the significance of the phenotypic difference between the mutants and wild-type plants.

For assessment the genetic interactions among *KRN4*, *UB2* and *UB3*, *krn4-mum* (referred to as *krn4*) was crossed with the *ub2-mum1 ub3-mum1* double mutant (referred to as *ub2/ub3*) in the W22 background as described by Chuck et al. (2014), followed by self-crossing to obtain the segregating $F_2$ population. A total of 577 individuals were phenotyped in the spring 2018 in Wuhan (30˚N, 114˚E) using gene-specific primers (S2 Table). For phenotyping, the widest diameter (*d1*) and the narrowest diameter (*d2*) at the tip of the ears were measured 20 days after pollination (DAP) (as shown in Fig 3A), and *d1/d2* was calculated to define ear fasciation. If d1/d2> = 1.2, the ear was considered fasciated. The frequency of ear fasciation was calculated from the number of fasciated ears to the total number of measured ears of a given genotype.

### Expression analysis

Total RNA was isolated from the tissues of ~2 mm and 5 mm ears of various maize plants (*KRN4*[3.1] and *KRN4*[1.9] in the H21 background, *krn4-mum* and W22 plants, *ub2::mum* and wild-type plants, *UB2* and *UB3* transgenic plants) using TRIzol reagent (Life Technologies, Invitrogen, Carlsbad, CA, USA) according to the manufacturer's instructions. cDNA was synthesized from mRNA using an EasyScript One-step gDNA-removal and cDNA-Synthesis Supermix Kit (Transgene, Beijing, China) according to the manufacturer's instructions. RT-PCR was conducted using gene-specific primers (S3 Table) and SYBR Green PCR Master

Mix (KAPA, Beijing, China). The maize *beta-actin* (*NM_001155179*) gene was used as an internal control. All assays were performed with three biological and three technical replicates.

### *In situ* hybridization

Immature B73 ears (2–3 mm) were fixed in a 4% PFA solution (4g of paraformaldehyde (Sigma-Aldrich) dissolved in 100 mL of 1× PBS, pH 6.5–7), embedded in paraplast plus (Sigma-Aldrich), and sectioned to a thickness of 8 μm. To construct sense and antisense RNA probes for *UB2*, *UB3*, *OBF1* and *OBF4*, probe fragments were amplified and cloned into pSPT18 (Roche), digested with *Hin*dIII and *Eco*RI, transcribed using SP6 and T7 RNA polymerase (Roche) *in vitro*, and labeled with digoxigenin-UTP (Roche). RNA hybridization, immunologic detection and signal capture of the hybridized probes were performed as described previously [51]. The *in situ* primers for *UB2*, *UB3*, *OBF1* and *OBF4* are listed in S3 Table.

### Dual-luciferase transient assay in maize protoplasts

To construct reporter plasmids, *KRN4^{3.1}*, *KRN4^{1.9}* and the *E1* segment were separately cloned into the site between *Bam*HI and *Sal*I, which is located approximately 2.8 kb from multiple cloning site (MCS) region, in the *PGL3-basic* vector (Promega, Madison, WI, USA) with *Minimal 35S* (*mp35S*)-driven *Luciferase* (*Luc*). The *UB3* promoter (*pUB3*) was also ligated to *mp35S* in the MCS region. To construct effector plasmids, the CDSs of *UB2*, *OBF1* and *OBF4* were individually cloned into the *pRTL2* vector with an enhanced *CaMV35S* promoter (*p35S*). The pRL-null vector (Promega) with *mp35S*-driven *Renilla Luciferase* (*REN*) was used as the internal control. The primer pairs used for vector construction are listed in S2 Table. Seedlings of maize inbred line B73 were grown in the dark at 28˚C for 13–15 days, and etiolated leaves were harvested for protoplast isolation. Transformation was performed as described previously [52], and dual-luciferase detection was conducted according to the manual of the Dual-LuciferaseReporter Assay System E1960 (Promega). For each assay, four to six biological replicates and two technological replicates were performed.

### Yeast one-hybrid assay

The full-length CDS of *OBF1* and *OBF4* were cloned into the *pGADT7* vector (Clontech, Beijing, China) to generate *pGADT7–OBF1* and *pGADT7–OBF4*, respectively. The *E1* segment was cloned into the *pHISi-1* vector (Clontech, Beijing, China) to generate *pHISi–E1*. The pHisi-E1 vector was linearized with *Xho*I and transferred into the yeast strain YM4271, followed by incubation for 3 to 5 days at 30˚C. A positive *pHISi-E1* clone was used to test the proper concentration of 3-amino-1, 2, 4-triazole (3-AT) in the SD/-His medium for background expression. Selection medium containing 20 mm 3-AT was used to screen for enhancer binding proteins. pGADT7, pGADT7–OBF1 and pGADT7–OBF4 were transferred into YM4271 yeast cells containing the *pHisi-E1* recombinant plasmid and grown in SD/-Leu-His supplemented with 20 mm 3-AT. After a 3 to 5–day incubation at 30˚C, positive clones confirmed by PCR-sequencing were diluted 10–100 fold and spotted onto SD/-Leu-His plates containing 0, 20 and 30 mm 3-AT. The primers used in the Y1H assay are listed in S2 Table.

### Electrophoretic mobility shift assay

To construct prokaryotic expression plasmids, the full-length CDS of *OBF1*, *OBF4* and *UB2* were cloned into *pMAL-c2x* (N–MBP) and *PGEX-4T-1* (N–GST), respectively. The expression of the OBF1–MBP, OBF4–MBP and UB2-GST fusion protein was induced in *Trans BL21* cells

and *Transetta* (DE3) *E. coli* cells (Transgene, Beijing, China) with 0.2 mM isopropyl-1-thio-D-galactopyranoside at 16˚C for 14 h. The OBF1–MBP and OBF4–MBP fusion proteins were purified using amylose resin (E8021S, NEB, USA), and the UB2-GST fusion proteins was purified using Glutathione Sepharose 4 EF (GE Healthcare Life Sciences China, Beijing, China), and the proteins were then quantified using the protein assay reagent (Sangon Biotech, Shanghai, China) following the manufacturer's protocol and detected by 12% SDS-PAGE. Single-stranded DNA probes were synthesized with a 5'-end biotin label (S1 Table); probe dilution, renaturation and binding reactions for the EMSA were performed as described previously [28]. The purified proteins (20–40 ng fusion protein per reaction) were incubated with labeled probes (final concentration 10 fM) in binding reaction mixture (10 mM Tris, 50 mM KCL, 1 mM dithiothreitol, 5 mM $MgCl_2$, 2.5% glycerol, poly (dI-dC), 0.05% NP-40; pH 7.5) at room temperature for 20 min. The reaction products were combined with protein loading buffer (25 mM Tris–HCL [pH 7.5], 0.04% bromophenol blue, 80% glycerol), then loaded onto 6% non-denaturing PAGE gels and subjected to electrophoresis (100 V) at 4˚C for 1–1.5 h in 0.5× TBE buffer (45 mM Tris, 45 mM boric acid, 2 mM EDTA, pH 8.3). After transfer to a membrane (100 V at 4˚C for 1 h in 0.5× TBE buffer) and UV crosslinking (120 mJ/cm$^2$, UV-light cross-linking instrument with 254 nm bulbs, 45–60 s exposure), shifts on the nylon membrane were detected using an LightShiftChemiluminescent EMSA Kit and Chemiluminescent Nucleic Acid Detection Module following the manufacturer's protocols (Thermo Scientific, Waltham, MA, USA).

## Chromatin immunoprecipitation and data analysis

The *pCaMV35S::UB2-YFP* construct was transformed into the maize inbred line ZZC01 Three transgenic overexpression lines, UB2–OE20, UB2–OE21 and UB2–OE23, were produced by CHINASEED (China National Seed Group Co. Ltd., Beijing, China). The transgenic and corresponding nontransgenic plants were grown in the field. Target gene expression was verified by fluorescence detection and immunoblot analysis (S2 Fig). Immature ears (5 mm long) were collected from UB2-OE20, dissected and cross-linked in 30 of mL 1× PBS buffer (137 mM NaCl, 2.7 mM KCl, 10 mM $Na_2HPO_4$, 2 mM $KH_2PO_4$) containing 1% formaldehyde for 15 min under a vacuum. Cross-linking was terminated by adding 0.15 M glycine, followed by 5 min of incubation under vacuum. The samples were washed three times with distilled water, dried with paper towels and quickly frozen in liquid nitrogen. Chromatin was extracted with extraction buffer 1 (EB1, 0.4 M sucrose, 10 mM Tris-HCl pH 8.0, 10 mM MgCl2, 5 mM mercaptoethanol and Plant Protease Inhibitor Cocktail). The sample was centrifuged for 20 min at 1000 g at 4˚C, and the pellet was washed five times with 5 mL EB2 (0.25 M sucrose, 10 mM Tris-HCl, pH 8.0, 10 mM $MgCl_2$, 1% Triton X-100, 5 mM -mercaptoethanol and plant protease inhibitor cocktail) and once with 5mL EB3 (1.7 M sucrose, 10 mM Tris-HCl, pH 8.0, 2 mM MgCl2, 0.15% Triton X-100, 5 mM -mercaptoethanol and a plant protease inhibitor cocktail), followed by sonication in 300 μL of sonication buffer (50 mM Tris-HCl, pH 8.0, 10 mM EDTA, 1% SDS and plant protease inhibitor cocktail) using a Diagenode Bioruptor sonicator (Diagenode SA, Seraing, Belgium) for 15 cycles (30 s ON/30 s OFF). The sonicated DNA fragments were 200–700 bp long. DiaMag protein A-coated magnetic beads (C03010020, Diagnode SA, Seraing, Belgium) were incubated with anti-green fluorescent protein (A11122, Thermo Fisher Scientific, Waltham, MA, USA), and the sonicated chromatin was immunoprecipitated with antibody-bead complexes. DNA elution, decrosslinking and isolation were performed using an IPure Kit v2 according to the manufacturer's instructions (C01010150, Diagenode SA, Seraing, Belgium). A small quantity of sonicated chromatin that was not incubated with the antibody was used as the total input DNA control. DNA libraries were

constructed using an Ovation Low Input DR Kit (NuGEN Technologies) and sequenced on the Illumina HiSeq 2500 platform, including two IP and two input libraries.

The FastQC program was used to assess high-quality clean reads [53], which were aligned to the maize reference genome (AGPv2) using Hisat2 v.2.0.5 [54]. Picard Mark Duplicates (v.2.9.0) and MACS (v.1.4.2) were used to remove PCR duplicates and for peak calling, respectively [55]. The *p*-value of the enriched peaks was $p < 1e-05$ in each ChIP-seq library compared to the input DNA. Integrative Genomics Viewer was used to visualize the ChIP tracks of the UB2-YFP fusion protein binding sites [56].

### ChIP–qPCR

ChIP-qPCR was performed with specific primers within targeted peaks located at *KRN4* and the *UB3* promoter region. The maize *tubulin* gene (AC195340.3) was used as the internal control. Two biological replicates and three technological replicates were performed. The primers used for quantitative ChIP–PCR analysis are listed in S3 Table. The abundance of a target was normalized to genomic regions with nonspecific binding genomic regions, and the fold enrichment was compared to the input sample. *P*-values were obtained via Student's t-test.

### Firefly luciferase complementation imaging

To analyze protein-protein interactions, the full-length CDS of *UB2*, *OBF1* and *OBF4* were cloned into JW771 (NLUC) and JW772 (CLUC) to produce recombinant the UB2–NLUC, OBF1–NLUC, OBF4–NLUC, UB2–CLUC, OBF1–CLUC and OBF4–CLUC vectors, respectively [57]. Young but fully expanded *Nicotiana benthamiana* leaves (8–10 weeks old) were infiltrated with *Agrobacterium tumefaciens* GV3101 cells that had been transfected with the above constructs. At four different areas on a leaf, one sample and three negative control combinations were infiltrated. After 48 h of growth under a 16 h-light/8 h-dark cycle, the abaxial epidermis of each leaf was coated with 1 mM luciferin (Promega). LUC signals were captured using the Tanon-5200 imaging system (Tanon, Shanghai, China). The experiments were repeated three to four times to obtain consistent results.

### Supporting information

**S1 Fig. Kernel row number in the *krn4*::*mum* mutant and wild-type plants.**
(A) Schematic diagram of the genomic locations of *KRN4* and *UB3* and the *Mutator* insertion site. Bar = 500 bp. Red line shows *KRN4* locus.
(B) Ears of wild type and *krn4-mum*. A represent ear of *krn4-mum* having 16 kernel rows (right) and a represent ear of wild type having 14 kernel rows (left) are shown. Bar = 4 cm.
(TIF)

**S2 Fig. Tassel phenotype of *UB3* and *UB2* transgenic lines.**
(A) Tassel length of the three independent *UB3* transgenic lines (UB3-UP) and non-transgenic lines (UB3-NT).
(B-D) Tassel branch and tassel length of *UB2* transgenic plants (UB2-OE) and non- transgenic plants (UB2-NT).
(E) Immunoblot analysis using YFP antibody. Proteins were extracted from the ears of UB2-OE20 and UB2-OE21 transgenic plants.
(F) UB2-YFP fusion protein signals observed in 2 mm ears by confocal microscopy. Strong signals indicate that UB2-YFP fusion protein was expressed in the ears of the transgenic lines.
 n, number of individual tassel. Values are means ± SD, *P*-value was calculated by Student's t-

test. *, p<0.05; **, p<0.01.
(TIF)

**S3 Fig. Transient assays of luciferase activity using *UB2* effector and *KRN4-UB3* promoter-driven reporter.**
(A) Effector and reporter constructs.
(B) Relative luciferase activity determined by transient expression analysis of *KRN4-UB3* promoter-driven *LUC* in maize protoplasts co-infiltrated with UB2 effector. For each transient assay, 4–6 biological replicates and two technical replicates were performed. The value is presented as mean (LUC/REN) ± SD. ** $P < 0.01$; * $P < 0.05$.
(TIF)

**S4 Fig. The spatial interaction loops formed in the KRN4—UB3 interval.**
(TIF)

**S5 Fig. The 84bp *E1* sequences in both *KRN4^{1.9}* and *KRN4^{3.1}*.** Sequences highlighted in red are enhancer-like elements. The sequence highlighted in green is the bZIP binding motif.
(TIF)

**S6 Fig. A putative complex for *KRN4* regulating *UB3* transcription in *cis*.** The two long-distant elements (*KRN4*, *UB3* promoter) kept in close proximity to each other by chromatin interaction. Three interacting transcription factors UB2, OBF1 and OBF4 bind to specific *cis*-elements (GTAC motif or enhancer elements) harbored in *UB3* promoter and *KRN4* to promote the transcription of *UB3* which negatively controls the initiation of reproductive axillary meristems and in turn the ear inflorescence branching.
(TIF)

**S1 Table. Probes used for the electrophoretic mobility shift assay.**
(DOCX)

**S2 Table. Primers used for vector construction and genotype identification.**
(DOCX)

**S3 Table. Primers used for quantitative RT-PCR and *in situ* hybridization.**
(DOCX)

## Author Contributions

**Data curation:** Yanfang Du, Yong Peng, Manfei Li, Zuxin Zhang.

**Formal analysis:** Yanfang Du, Lei Liu, Manfei Li.

**Funding acquisition:** Zuxin Zhang.

**Methodology:** Yanfang Du, Yong Peng, Yunfu Li, Dan Liu, Xingwang Li, Zuxin Zhang.

**Project administration:** Zuxin Zhang.

**Software:** Yanfang Du.

**Supervision:** Yanfang Du, Lei Liu.

**Writing – original draft:** Yanfang Du.

**Writing – review & editing:** Yanfang Du, Zuxin Zhang.

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
