## [Decision Letter · Decision Letter 0]

8 Feb 2020

Dear Dr Zhang,

Thank you very much for submitting your Research Article entitled 'UNBRANCHED3 Expression and Inflorescence Development is Mediated by UNBRANCHED2 and the Distal Enhancer, KRN4, in Maize' to PLOS Genetics. Your manuscript was fully evaluated at the editorial level and by independent peer reviewers. The reviewers appreciated the attention to an important topic but identified some aspects of the manuscript that should be improved.

We therefore ask you to modify the manuscript according to the review recommendations before we can consider your manuscript for acceptance. Your revisions should address the specific points made by each reviewer.

[LINK]

Yours sincerely,

Sarah Hake

Associate Editor

PLOS Genetics

Gregory P. Copenhaver

Editor-in-Chief

PLOS Genetics

The reviewers have suggested mostly changes to writing, statistical issues, figure legends, etc. but reviewer 1 is concerned about controls for one experiment which might require additional experimental work. Please address all their concerns completely.

Reviewer's Responses to Questions

**Comments to the Authors:**

Reviewer #1: Maize kernel row number is an important agronomical trait that influences grain yield. The genes UB2, UB3 and KRN4 locus are key components regulating kernel row number and inflorescence development. Some previous studies have indicated that these components genetically interact with each other. However, it is still unclear how these components work together to regulate inflorescence development. In the present manuscript, Du et al. investigated the genetic and molecular effects of KRN4 with UB2 and UB3 to reveal the mechanism whereby KRN4 fine-tunes UB3 expression and, thus, inflorescence development. This work is of interest for the readership and is suitable to be published in PLOS Genetics after minor revision.

Major concerns:

1. On the figure 6D (line 683), the authors used Reporter-CK construct as control, that's very inappropriate! They should use “Reporter-E1” as control and use “Reporter-E1 + Effector-OBF1 or Effector-OBF4” as experimental group, because as shown in Fig 5B, Reporter-E1 itself shows much higher LUC activity than Reporter-CK.

The same problem occurs in Fig S3 (line 728).

2. The results in the manuscript show that UB2, OBF1 and OBF4 can directly bind to KRN4 region (Fig 4G and 6A-B), and each of them can elevate the KRN4-mediated enhancement effect on downstream gene expression (Fig 6D and S3). It is of interest whether these three TFs show synergetic effect on KRN4-mediated expression of downstream genes, such as UB3. I suggest the authors investigate this synergetic effect by the combination of three TFs using transient expression assays.

Minor concerns:

1. Fig 6A, the yeast clones growing on 0 mM 3-AT plate are smaller than those in 20 mM 3-AT and 30 mM 3-AT, a scale bar needs to be added.

Fig 6B, the shifted bands are not in line with the free probes, it seems that they are not derived from the same membrane.

2. Line 696, “KRN4 binding proteins”, instead of “KRN4 binging proteins”.

3. Line 755, Table S1, it’s better to underline the mutant sites in 3×E1 repeat, KRN4-P4 and pUB3-P1 sequences.

Reviewer #2: The authors previously have revealed that KRN4 controls quantitative variation in maize kernel row number, and UNBRANCHED3 regulates branching by modulating cytokinin biosynthesis and signalling in maize. In the present study, the authors provide experimental evidence for genetic interactions between KRN4 and UB3 as well as UB2 in controlling inflorescence architecture, enhancer activity of KRN4 toward UB3 promoters, and KRN4 recruited UB2-centred transcription complex for UB3 transcription. The findings provide new insights for improving KRN and yield in maize. The authors need address the following points before it could be accepted for publishing in PLoS Genetics.

1. The manuscript needs thoroughly language editing before accepting for publication.

2. Line 165, UB2 transcription “levels” exhibited >8-fold increases, “and” UB3…

3. Line 351, should be “Total RNA” while “mRNA” in Line 355

4. Line 618, the reference is not properly numbered.

5. Figure 1,

a) Please specify what the red lines stand for

b) Figure legends required for the bar charts

c) Is the expression level calculated in relative to blue bars? Please clarify.

d) No significance at P < 0.05 level so the figure legends to be rephased

6. Figure 2

a) Again, how is the expression data calculated or normalised? Error bar means SD or SE?

b) Is inflorescence length and number affected by UB3 expression?

c) In the subpanel H, please clarify the images with enlarged zones and add annotations to the red arrows.

7. Figure 3

a) Why the ratio is set at 1.2 to define a fascinated ear? What is the rate of the representative cone in A? The image would cause misleading if the value is not 1.2.

b) A comparison of cross sections of the ear tips among mutants will be more intuitionistic.

c) The authors proposed the increased proportion of fascinated ears in krn4;ub2 is associated with the significant inhibition of UB3 transcription. The results showed a coordinated role of KRN4 and UB2 in the regulation of UB3 transcription, however, the fascination frequency could not match the UB3 expression since the krn4;ub2;UB3 had the highest rate but not lowest gene expression. Please explain.

8. Figure 4

a) It seems that the tassel size is significantly reduced in the OE lines, thus any phenotyping is conducted?

b) Please add colour legend to D.

9. Figure 6, after how many days the yeast culture is photographed? The OBF1 binding activity seems very low as observed from the colony. Please explain.

10. Figure 7, the figure has missing subpanels according to the legends so please add the relevant images. Line 702, should be “NP and CP” but not PN and PC.

11. 'L34 A large proportion of the plant genome consists of intergenic regions that were previously thought to lack biological functions'. This is an outdated statement and should not be in the Author summary

Reviewer #3: Du et al. described a series of experiments to investigate the regulation of UB3 expression by KRN4, which is an intergenic region ~60 kb away. I feel that all relevant genetic materials were carefully generated, and experiments were done appropriately. The topic is relatively new, and the overall study is novel. The findings from this mechanistic study are very interesting to a broad range of scientists. I enjoyed reading the manuscript.

Only some minor comments.

1. The current discussion is narrow. It may be good to add a few general statements in Discussion to explain that “cis-regulatory element” still be “long-distance” and “distal”, similar to what you have in L81-86. And a brief discussion of other known cases?

2. Authors may discuss the genetic effect of the original mapping of KRN4 for kernel row number and the current findings. It is known that materials used for mechanistic study are often different from the mapping study. But with the currently proposed model, it would be interesting to try to explain the situation in the original mapping and cloning study.

3. I found a few issues in Fig. S1, which was supposed to support Fig. 1. While authors showed the picture of kernel row number of 14 for wild type and 16 for krn4-mum, the actual average numbers were roughly 14 and 14.5(?). Yes, it is significantly different, but authors need to point out that what was shown for krn4-mum (16) can only be observed from some of the ears. Authors need to point out this in the main text too.

Fig. S1C, something must be wrong for the right two bars (Sanya), if we believe what is show on the left two bars (Wuhan) is correct. It is not possible to obtain a p=0.03 if the standard deviation bars are so wide and the sample size (87/88) is only slightly higher than the left two bars (Wuhan, 76/84). On the other hand, it is likely that the left two bars are wrong, so are the Fig. 1F. The standard deviation bars should be wider that what is shown if you have row numbers of 12, 14, and 16. If authors figured out it is not significantly different for Sanya, this needs to be noted in Fig. S1, since only the results from Wuhan is in Fig. 1F.

In Fig. 1 “One-way ANOVA” was mentioned, but “Student’s t-test” was mentioned in Fig. S1. Please be consistent in description and analysis. Regardless whether these two terms mean the same thing in your case or give the same results. Check all other places.

Please inspect the raw data and analyses to make sure the calculations and plotting are all correct. Also, reviewing the difference between standard deviation and standard error would be helpful.

Please revise so that the color scheme is consistent with Fig. 1. For Fig. S1A, either remove the label “krn4-mum”, or move it up a bit to avoid confusion.

**Have all data underlying the figures and results presented in the manuscript been provided?**

Reviewer #1: None

Reviewer #2: Yes

Reviewer #3: None

PLOS authors have the option to publish the peer review history of their article (what does this mean?). If published, this will include your full peer review and any attached files.

Reviewer #1: No

Reviewer #2: No

Reviewer #3: No

---

## [Decision Letter · Decision Letter 1]

7 Apr 2020

Dear Dr Zhang,

We are pleased to inform you that your manuscript entitled "UNBRANCHED3 Expression and Inflorescence Development is Mediated by UNBRANCHED2 and the Distal Enhancer, KRN4, in Maize" has been editorially accepted for publication in PLOS Genetics. Congratulations!

Yours sincerely,

Sarah Hake

Associate Editor

PLOS Genetics

Gregory P. Copenhaver

Editor-in-Chief

PLOS Genetics

Comments from the reviewers (if applicable):

Dear Dr. Zhang,

Two of the three reviewers are happy with the changes you made. I interacted with the third reviewer by email and he/she was also satisfied with your revision. I am happy to write that the manuscript is now accepted.

Reviewer's Responses to Questions

**Comments to the Authors:**

Reviewer #1: The authors have carefully considered my comments on the original manuscript. The revised manuscript is suitable to be published in PLoS Genetics.

Reviewer #3: I noticed the changes author made. No additional comments.

**Have all data underlying the figures and results presented in the manuscript been provided?**

Reviewer #1: None

Reviewer #3: Yes

PLOS authors have the option to publish the peer review history of their article (what does this mean?). If published, this will include your full peer review and any attached files.

Reviewer #1: No

Reviewer #3: No

**Data Deposition**

http://datadryad.org/submit?journalID=pgenetics&manu=PGENETICS-D-20-00024R1

**Press Queries**

---

## [Editor Report · Acceptance letter]

17 Apr 2020

PGENETICS-D-20-00024R1 

UNBRANCHED3 Expression and Inflorescence Development is Mediated by UNBRANCHED2 and the Distal Enhancer, KRN4, in Maize 

Dear Dr Zhang, 

We are pleased to inform you that your manuscript entitled "UNBRANCHED3 Expression and Inflorescence Development is Mediated by UNBRANCHED2 and the Distal Enhancer, KRN4, in Maize" has been formally accepted for publication in PLOS Genetics! Your manuscript is now with our production department and you will be notified of the publication date in due course.

With kind regards,

Matt Lyles

PLOS Genetics

On behalf of:
